# Androgen-Induced MIG6 Regulates Phosphorylation of Retinoblastoma Protein and AKT to Counteract Non-Genomic AR Signaling in Prostate Cancer Cells

**DOI:** 10.3390/biom12081048

**Published:** 2022-07-29

**Authors:** Tim Schomann, Kimia Mirzakhani, Julia Kallenbach, Jing Lu, Seyed Mohammad Mahdi Rasa, Francesco Neri, Aria Baniahmad

**Affiliations:** 1Institute of Human Genetics, Jena University Hospital, Am Klinikum 1, 07740 Jena, Germany; timmeeh@gmx.de (T.S.); kimiamirzakhani@rocketmail.com (K.M.); julia.kallenbach@uni-jena.de (J.K.); 2Leibniz Institute on Aging, Beutenbergstraße 11, 07745 Jena, Germany; jing.lu@leibniz-fli.de (J.L.); mahdi.rasa@leibniz-fli.de (S.M.M.R.); francesco.neri@leibniz-fli.de (F.N.)

**Keywords:** prostate cancer, cellular senescence, bipolar androgen therapy, AKT signaling

## Abstract

The bipolar androgen therapy (BAT) includes the treatment of prostate cancer (PCa) patients with supraphysiological androgen level (SAL). Interestingly, SAL induces cell senescence in PCa cell lines as well as ex vivo in tumor samples of patients. The SAL-mediated cell senescence was shown to be androgen receptor (AR)-dependent and mediated in part by non-genomic AKT signaling. RNA-seq analyses compared with and without SAL treatment as well as by AKT inhibition (AKTi) revealed a specific transcriptome landscape. Comparing the top 100 genes similarly regulated by SAL in two human PCa cell lines that undergo cell senescence and being counteracted by AKTi revealed 33 commonly regulated genes. One gene, ERBB receptor feedback inhibitor 1 (*ERRFI1*), encodes the mitogen-inducible gene 6 (MIG6) that is potently upregulated by SAL, whereas the combinatory treatment of SAL with AKTi reverses the SAL-mediated upregulation. Functionally, knockdown of *ERRFI1* enhances the pro-survival AKT pathway by enhancing phosphorylation of AKT and the downstream AKT target S6, whereas the phospho-retinoblastoma (pRb) protein levels were decreased. Further, the expression of the cell cycle inhibitor p15^INK4b^ is enhanced by SAL and *ERRFI1* knockdown. In line with this, cell senescence is induced by *ERRFI1* knockdown and is enhanced slightly further by SAL. Treatment of SAL in the *ERRFI1* knockdown background enhances phosphorylation of both AKT and S6 whereas pRb becomes hypophosphorylated. Interestingly, the *ERRFI1* knockdown does not reduce AR protein levels or AR target gene expression, suggesting that MIG6 does not interfere with genomic signaling of AR but represses androgen-induced cell senescence and might therefore counteract SAL-induced signaling. The findings indicate that SAL treatment, used in BAT, upregulates MIG6, which inactivates both pRb and the pro-survival AKT signaling. This indicates a novel negative feedback loop integrating genomic and non-genomic AR signaling.

## 1. Introduction

Prostate cancer (PCa) is an important age-related disease with the highest estimated incidence of new cancer cases [1]. Second only to lung cancer, it is one of the leading causes of cancer mortality in men in Western countries. The evidence shows that growth of the normal prostate tissue as well as the initial development of PCa relies on the activation of the androgen receptor (AR) [2]. Thus, the AR represents a major drug target in the treatment of PCa [3].

To inhibit AR signaling androgen deprivation therapy (ADT) and full blockade by AR antagonists are the major forms of PCa hormone therapy recommended for advanced and metastasized PCa. The growth inhibition of PCa by AR antagonists is associated with induction of cellular senescence [4,5,6,7]. Interestingly, also at supraphysiological androgen level (SAL), the proliferation of PCa is inhibited and cellular senescence is induced, an irreversible cell cycle arrest [7,8,9]. Accordingly, cellular senescence is induced in PCa samples from patients treated ex vivo [8]. SAL are used in clinical trials with the so-called bipolar androgen therapy (BAT) for treatment of metastatic PCa patients [10,11]. With BAT treatment, PCa patients receive intermittent androgen injections at doses shown to produce a spike in serum androgens to supraphysiological levels, followed by a decline to below castrate levels at the end of a 28-day treatment cycle. Rapid cycling of androgens from SAL levels to below normal levels may delay adaptive changes in AR signaling and thereby delaying the emergence of resistance [12].

Using a mouse model, it was further confirmed that the functional AR exhibits both proliferation-promoting as well as tumor suppressive functions [13]. Consistently, it was discovered that treatment with 1 pM R1881, defined as low androgen levels (LAL), increased proliferation of LNCaP cells, whereas treatment with 1 nM R1881, defined as SAL reduces proliferation and induces cellular senescence in LNCaP cells [8]. In contrast to dihydrotestosterone, which is rapidly metabolized and its metabolites may act as estrogen receptor beta agonists, the much less metabolizable synthetic androgen methyltrienolone (R1881) and thus more AR-specific androgen was used. Similar results were obtained using dihydrotestosterone [8]. Furthermore, SAL treatment hyperphosphorylates AKT, induces p16^INK4A^ and p15^INK4b^, reduces Cyclin D1, hypophosphorylates retinoblastoma protein (pRb) and enhances autophagy activity [8,9]. Using the AKT inhibitor (AKTi), the SAL-induced level of senescent cells was reduced in both castration-sensitive (LNCaP) and castration-resistant (C4-2) PCa cell lines [8,9]. The AR also has non-genomic activity in addition to its genomic activity as a transcription factor. This includes the phosphorylation of AKT and activation of AKT downstream signaling through mTOR and S6 by SAL [8]. Interestingly, the AR-AKT interaction has been shown previously and indicates that SAL mediates cell senescence in part by this non-genomic androgen signaling through the AR-AKT pathway. 

In this study, we induced cell senescence in the two human PCa cell lines, LNCaP and C4-2, and performed RNA-seq. In addition, we performed transcriptome analysis from data of both cell lines treated with SAL, AKTi alone and AKTi in combination with SAL to identify genes that are upregulated by SAL and downregulated by AKTi in order to define genes in the SAL-AKT pathway. These factors may function as pro- or as anti-senescence factors in the SAL-AKT-mediated pathway to induce cell senescence. We identified and characterized MIG6 as a factor strongly regulated by AR and AKT. Since the knockdown of MIG6 reveals similar changes in the AKT-S6 and p15-pRb pathway as SAL, it suggests that MIG6 is involved in a negative feedback loop of AR-SAL signaling. The data indicate that MIG6 regulates pRb and AKT phosphorylation and might be a modulator of the SAL-induced non-genomic pathway.

## 2. Materials and Methods

### 2.1. Cell Culture and Treatments

The LNCaP (lymph node prostate cancer) cell line [14] was used as a model of human androgen-dependent PCa. Cells were cultured in RPMI 1640 medium (Gibco Life Technologies, Carlsbad, California, CA, U.S.) supplemented with 5% fetal calf serum (FCS), penicillin (100 U/mL), streptomycin (100 µg/mL), 1% sodium pyruvate and 25 mM of HEPES pH 7.5 (Carl Roth, Karlsruhe, Germany). The C4-2 cell line is a derivative of LNCaP and was used to represent castrate-resistant PCa (CRPC) cells. Comparisons between these cell lines may reveal common and distinct signaling representing castration sensitivity and resistance. Cells were cultured in DMEM supplemented with 20% F12, 5% FCS, penicillin (100 U/mL), streptomycin (100 µg/mL), 1% sodium pyruvate and 25 mM of HEPES pH 7.5. All cells were maintained in a 5% CO_2_, humidified atmosphere at 37 °C (Thermo Fisher Scientific, Waltham, Massachusetts, MA, U.S.). Both cell lines were seeded in an appropriate amount for each experiment in cell culture plates. After 48 h of incubation, the cells were treated for 72 h with 1 pM R1881 (LAL), 1 nM R1881 (SAL), 10 µM bicalutamide (Bic), 10 μM enzalutamide (Enz) or 0.1% DMSO (Carl Roth, Karlsruhe, Germany) as solvent control (C) in a 5% CO_2_, humidified atmosphere at 37 °C.

### 2.2. RNA-Sequencing and Transcriptome Analysis 

RNA-sequencing and transcriptome analysis of both LNCaP and C4-2 cell lines were previously described [9]. Cells were treated for 3 days with DMSO and the AR-specific agonist R1881, with and without the AKT inhibitor (1 µM) AKTi, prior to RNA isolation. The RNA-sequencing data is available in the gene expression omnibus (GEO) database under the accession numbers GSE162711, GSE155528, and GSE154755.

### 2.3. Cellular Senescence Assays

The assays were performed with 6-well plates, and the cells were seeded at 25,000 cells per well. The staining and detection were performed as described previously [15,16,17]. The percentage of SA-β-Gal positive cells was calculated by counting at least 3 × 200 cells per well and at least 2 wells per treatment under a light microscope (Zeiss, Oberkochen, Germany).

### 2.4. Antibodies and Western Blotting

For protein extraction, the assays were performed with 10-cm cell culture plates, and the cells were seeded at 500,000 cells per dish. Briefly, the cells were lysed using NETN buffer [8] supplemented with phosphatase inhibitors [8] and followed by three cycles of freezing (in liquid nitrogen) and thawing (in a 37 °C water bath). The protein extracts were separated by SDS-PAGE. The primary antibodies used for immunodetection were for MIG6 (Proteintech, 11630-1-AP), panAKT (Cell Signaling, 4685S, lot 6), AR (Biogenex, 256M), β-Actin (Abcam, ab6276, GR3324554-1), p-AKT (S473) (Cell Signaling, 4058S, lot 14), pRb (Abcam, ab6075, lot 821737), phospho-pRb (Cell Signaling, 9308, lot 13), panS6 (Cell Signaling, 2217S, lot 10), and p-S6 (S235/236) (Cell Signaling, 2211S, lot 23). Horseradish peroxidase-conjugated anti-mouse IgG (Cell Signaling, 7076S, lot 32) or anti-rabbit IgG (Cell Signaling, 7074S, lot 28) were used as secondary antibodies. The detection was performed by ImageQuant^TM^ LAS 4000 (GE Healthcare Bio-Sciences AB, Chicago, Illinois, U.S.). Quantification of bands was performed via the LabImage D1 program.

### 2.5. Immunofluorescence Staining

LNCaP cells were seeded in RPMI 1640 medium containing 5% normal untreated FBS and cultured for 48 h. After 72 h of ligand treatment, cells were fixed with 4% paraformaldehyde and permeabilized with 0.25% Triton-X100/PBS [9] for 10 min at room temperature. After three washing steps in 1x PBS, a blocking solution (5% Normal Goat Serum/PBS) was added for 1 h. Primary antibodies were incubated in a humidified chamber overnight at 4 °C. Goat anti-rabbit secondary antibodies were incubated for 1 h at room temperature. After washing, cells were stained with Hoechst in 1x PBS followed by mounting with Flouremount G. Images were obtained with a confocal laser scanning microscope (Zeiss LSM 880, Oberkochen, Germany) with Airyscan in super resolution using a Plan-Apochromat 63x/1.4 oil DIC M27 objective confocal scanning fluorescence microscope. Fiji software (2.5.0) (https://fiji.sc/ Accessed on 20 January 2022) was used for analysis of the images.

### 2.6. Quantitative Reverse Transcription PCR (qRT-PCR)

The assays were performed with RNA isolated from 10-cm cell culture dishes, and the cells were seeded at 500,000 cells per dish. The total RNA extraction was performed using peqGOLD TriFast (Peqlab, Erlangen, Germany) according to the manufacturer’s protocol. Two-step qRT-PCR was performed as described previously [16,17] with gene-specific primers. *TBP* and *GAPDH* mRNA served as the housekeeping gene for normalization. The primer sequences are listed (Table 1) as 5′→3′:

### 2.7. ChIP-Seq and ChIP-Seq Analysis

Chromatin immunoprecipitation (ChIP) was performed according to the manufacturer’s protocol (iDeal ChIP-seq Kit Diagenode, Cat.-Nr.: C01010055, Denville, U.S.). Cells were harvested and cross-linked with ChIP cross-link Gold (Diagenode,) for 30 min and 1% formaldehyde for 10 min followed by quenching with glycine. Chromatin was isolated and sheared using Bioruptor Pico for 4 cycles (30 s on, 30 s off). Sheared and digested chromatin was verified on agarose gels to obtain 150–900 bp fragments following incubation with antibody-bound protein A conjugated beads overnight at 4 °C with rotation. For immunoprecipitation, antibodies against AR (Cell signaling) and AKT (ThermoFisher, Waltham, Massachusetts, U.S.) were used. IgG-coupled beads served as negative control. 1% input is indicated. ChIPed DNA was eluted from beads, purified by IPure magnetic beads and then used for ChIP-qPCR or ChIP-seq library preparation. Preparation of ChIP-Seq libraries (TruSeq ChIP-seq) and ChIP-sequencing were performed by Macrogen (Seoul, South Korea). Sequencing was performed using the NovaSeq 6000 platform at a 2 × 150 bp configuration and with an output of 5 Gb (30 Mio reads) per sample.

Quality control was performed by “fastqc (v0.11.9, Simon Andrews, released in Babraham institute, Cambridgeshire England, available online at: http://www.bioinformatics.babraham.ac.uk/projects/fastqc/) (Accessed on 20 January 2022)” and “fastqscreen (v0.14.0, Steven Wingett, released in Babraham institute, Cambridgeshire, England, available online at: https://www.bioinformatics.babraham.ac.uk/projects/fastq_screen/) (Accessed on 20 January 2022)” software. Fastq files were aligned to reference genome (GRch37/hg19) using Bowtie (v1.1.2, Ben Langmead, College Park, MD, USA) with parameters as “bowtie -q --max/dev/null --chunkmbs 3,200,000 -p 4 -S --sam-nohead --best --strata -m 1”. Then, SAM files were sorted by genomic coordinates in commands of “sort -k 3,3 -k 4,4n -T”. Reads in SAM files are marked for duplication and kept with the initial length of 150 bp. Software MACS v1.4 tool (Yong Zhang, Boston, MA, USA) was used to generate wiggle files from sorted SAM files, by parameters of “macs14 --gsize=hg19 -t input_file --format=sam -w -S -p 1e-5 --nolambda --nomodel -n output_file”. Igvtools (v2.3.97, James T. Robinson, Cambridge, MA, USA) was applied to transfer the wiggle file into a tdf file. Peaks are called in treatment samples with paired control samples by software MACS (v2.2.6, Young Zhang, Boston, MA, USA) with parameters of “macs2 callpeak -t treatment_sample -c control_sample -f SAM -g hs -p 1e-5 --slocal 1000 --llocal 10,000 -n output_file”. Peaks were annotated for promoters, exons, introns and intergenic regions by the script “annotatePeaks.pl” in software “HOMER” (v4.9.1, Sven Heinz, San Diego, CA, USA).

### 2.8. Statistical Analysis

For statistical analysis a two-tailed unpaired Student’s *t*-test was performed using the GraphPad Prism 8.0 software, which was calculated from the mean, standard deviation (SD), standard error of mean (SEM) and number of replicates (n). A 95% confidence interval (*p*-value (*p*) < 0.05) was considered as statistically significant (*) between two subject groups. A 99.5% confidence interval (*p* < 0.05), 99% confidence interval (*p* < 0.01) and a 99.9% confidence interval (*p* < 0.001) were indicated by one (*) two (**) and three stars (***), respectively. A 99.99% confidence interval (*p* < 0.0001) was indicated by four stars (****). Western blotting analysis was performed for at least three biological replicates.

## 3. Results

### 3.1. RNA-seq Identifies ERRFI1 Being Upregulated by SAL in Both LNCaP and C4-2 Cells

RNA-seq was performed and analyzed for an overlap of androgen-mediated upregulated genes in two cell lines, the androgen-dependent LNCaP and the castration-resistant C4-2 cell lines. In contrast to LAL treatment, SAL induces cellular senescence in both cell lines [9]. Interestingly, inhibition of AKT by AKTi represses androgen-induced cellular senescence. The top 100 genes with high score being specifically upregulated by SAL in each cell line were further analyzed for their common upregulation in both cell lines leading to 33 genes (Figure 1A and Appendix A). 

One of the prominently induced genes is *ERRFI1*, encoding MIG6, being upregulated significantly by SAL, whereas LAL did not upregulate *ERRFI1* expression (Figure 1B–D). AKTi treatment itself did not affect the basal mRNA expression of *ERRFI1* (Figure 1E). Since the combination treatment of SAL with AKTi did not show statistically significant downregulation, qRT-PCR experiments were used for both cell lines. The upregulation of *ERRFI1* by SAL was confirmed and the downregulation of *ERRFI1* by AKTi was revealed by qRT-PCR in both cell lines (Figure 2A,B). Notably, treatment with first and second-generation AR antagonists, enzalutamide and bicalutamide, did not induce the expression level of *ERRFI1* (Appendix A). To verify an upregulation of MIG6 at protein level, both cell lines were treated with LAL or SAL in the absence or presence of AKTi (Figure 2C). The data indicate an upregulation of MIG6 at protein level by SAL treatment in both cell lines, whereas AKTi reduces MIG6 levels to a different extent comparing LNCaP with C4-2 cells. ChIP-seq experiments indicate the recruitment of AR up- and downstream of the *ERRFI1* locus (Figure 2D). More reads were obtained at this locus by SAL treatment. The data suggest that *ERRFI1* is a direct AR target gene. 

The dose-dependent treatment with androgen indicates that *ERRFI1* mRNA expression is induced at 1 nM R1881 (Appendix A). Time-dependent androgen treatment indicates that MIG6 protein levels are enhanced at 48 and 72 h treatment with SAL (Appendix A). In contrast, treatment with the AR antagonist Enz did not upregulate MIG6 protein levels at any time point.

### 3.2. SAL Treatment Enhances Cytosolic MIG6 Levels

In order to detect intracellular levels of MIG6 and the intracellular distribution of MIG6 in the absence or presence of SAL, high-resolution laser scanning microscopy was performed detecting endogenous expressed MIG6. LNCaP cells were cultured in normal serum and treated with or without SAL. Enhanced fluorescence signals were detected in the cytoplasm of LNCaP cells upon SAL treatment (Figure 3). These data indicate that androgens induce the expression of MIG6, which is preferentially localized in the cytoplasm.

### 3.3. AKTi Reduces SAL-Enhanced p-AKT and p-S6 Levels

The increase of phosphorylation of AKT (p-AKT) at serine 473 (S473) by SAL treatment in LNCaP cells was shown previously [8]. Similarly, as shown before, LAL had no effect on the levels of p-AKT. To confirm that AKTi inhibits AKT signaling in both LNCaP and C4-2 cells, the AKT downstream target S6 was analyzed. SAL enhances in both cell lines the levels of p-AKT and p-S6 (Figure 4) indicating non-genomic activity of AR signaling by SAL. Interestingly, AKTi inhibits p-AKT levels in LNCaP more pronounced compared to C4-2 level, which might be one basis of castration-resistance. In contrast, p-S6 levels were more reduced in C4-2 cells compared to LNCaP (Figure 4). This indicates that the AR interacts not only with AKT but also directly or indirectly with S6 kinase(s) to regulate their activity in the presence of AKT inhibition.

### 3.4. Knockdown of ERRFI1 Increases SA-β-Gal Positive Cells

SAL is known to activate the AKT-mTOR pro-survival pathway [8]. To analyze whether MIG6 is also involved in or mediates the SAL-induced phosphorylation of AKT and S6, knockdown experiments of *ERRFI1* were performed. According to the RNA-seq and qRT-PCR data, MIG6 encoded by *ERRFI1* is induced by SAL and downregulated by AKTi in the presence of SAL. Since SAL enhances phospho-AKT (p-AKT) levels, the hypothesis was to analyze whether MIG6 also regulates p-AKT levels. To verify this hypothesis, siRNA pool for *ERRFI1* and a non-targeting control pool (siControl), were transfected into LNCaP cells. The transfected LNCaP cells were treated for 72 h with SAL or DMSO as solvent control. *siERRFI1* significantly reduced SAL-induced mRNA level of *ERRFI1* (Figure 5A). At the protein level, the knockdown of MIG6 was confirmed by Western blotting using siRNA or short hairpin-mediated knockdown (Figure 5B,C). Of note, we did not detect upon knockdown of MIG6 an influence on AR protein level (Appendix A) or the expression of the direct AR target gene *FKBP5* (Appendix A).

Since enhanced p-AKT and hypo-phosphorylated pRb levels are linked to the AR mediated induction of cellular senescence by SAL in PCa cells [8,9], we analyzed whether MIG6 is involved in SAL-mediated cellular senescence. Knockdown cells using *siERRFI1* or *shERRFI1* were generated (Figure 5A–C). Interestingly, *siERRFI1* or *shERRFI1* transfected cells showed significantly higher basal SA-β-Gal positive cells compared to control transfected cells (Figure 5D and Appendix A with si-mediated knockdown), indicating that MIG6 protects PCa cells to undergo cellular senescence. Still, SAL treatment enhances further cellular senescence with a slightly but significantly higher percentage of SA-β-Gal positive cells in *ERRFI1*-knockdown cells. It should be taken into account that the fold-enhancement of SA-β-Gal positive cell level in the *ERRFI1*-knockdown cells is reduced. This may indicate that *ERRFI1* is involved in protecting these cancer cells to undergo cellular senescence by SAL.

### 3.5. ERRFI1 Knockdown Represses CCND1 and Induces the Expression of CDKN2B/p15^INK4b^

SAL-mediated cellular senescence is associated with hypophosphorylation of pRb as well as decrease of the E2F target gene *CCND1*, encoding Cyclin D1. To investigate whether MIG6 is involved in this pathway, phosphorylation levels of pRb and *CCND1* expression were analyzed in the knockdown background. The *ERRFI1* knockdown leads to reduced pRb levels, which were further decreased in the presence of SAL (Figure 6). In line with this and SAL-mediated induction of cellular senescence, the p-AKT and p-S6 levels were enhanced by *ERRFI1* knockdown and further induced by SAL treatment (Figure 6). 

These data confirm induction of cellular senescence by *ERRFI1* knockdown using a similar pathway as the AR. In accordance with that, *CCND1* expression was repressed, and this effect was further enhanced in the SAL treated *ERRFI1* knockdown cells (Figure 7A). In addition, the expression of the cell cycle inhibitor *CDKN1A*, encoding p21, shows an expression pattern being in line with the induction of cellular senescence by knockdown. The expression further increased by the combination of SAL and *ERRFI1* knockdown (Figure 7B). Recently, it was shown that the cell cycle inhibitor p15^INK4b^ mediates SAL-induced cellular senescence [9]. To analyze whether MIG6 also regulates the expression of this factor, knockdown cells were analyzed. Upon SAL treatment, the expression of *CDKN2B* mRNA, encoding p15^INK4b^, was enhanced and further upregulated in the *ERRFI1* knockdown cells (Figure 7C). Accordingly, at protein level p15^INK4b^ is potently enhanced in the SAL-treated knockdown cells (Figure 7D). 

Taken together, these data indicate that MIG6 regulates the key factors associated with and mediating induction of cellular senescence by SAL treatment. 

## 4. Discussion and Conclusions

SAL treatment induces cellular senescence in PCa cells. Treatment with AKTi abrogates SAL-induced cellular senescence, indicating that SAL-induced cellular senescence is mediated in part by non-genomic signaling. SAL treatment induces AR-AKT signaling pathway. MIG6 is known as the mitogen-inducible gene and is mainly known for its feedback inhibitor function of ERBB-2 mitogenic and transforming signals indicating oncogenic function [18,19,20]. However, MIG6 also exhibits tumor suppressor activity presumably in a cell type and context-dependent manner [21,22,23]. In line with this, both MIG6-mediated inhibition and stimulation of AKT phosphorylation were reported [24]. In endometrial epithelial cells MIG6 suppresses proliferation by inhibiting p-AKT [25]. MIG6 expression decreased migration and invasion of MEK-inhibited mutant NRAS melanoma, especially in response to epidermal growth factor stimulation, also indicating a tumor suppressive role in other cancers [26].

Since inhibition of AKT reduced SAL-mediated induction of cellular senescence in PCa cells, it indicates that AKT-signaling pathway is partly essential for SAL-mediated induction of cellular senescence [8,9]. Interestingly, the knockdown of *ERRFI1* induces cellular senescence as well as the levels of p-AKT and downstream target p-S6, suggesting an activation of AKT-signaling by knockdown. This is in line with the suggestion that SAL increases p-AKT levels and downstream AKT signaling in order to enhance a pro-survival pathway and to induce cellular senescence [7,27]. The knockdown of *ERRFI1* further increased SAL-mediated induction of cellular senescence but reduced the fold induction of cellular senescence compared to control transfected cells. The knockdown experiments suggest that MIG6 inhibits AKT phosphorylation and represses the expression of p15^INK4b^, in order to suppress the induction of cellular senescence by SAL. This may suggest that the knockdown of ERRFl1 causes attenuation of the increased number of SA-Gal positive cells when compared to the increase upon SAL treatment in shLuc cells, and therefore, MIG6 might be involved in the SAL inhibition of tumor progression. These findings also suggest that SAL treatment enhances in addition to p-AKT levels, MIG6 expression. MIG6 subsequently represses phosphorylation of AKT. The data suggest that SAL treatment induces, through MIG6, a negative feed-back loop to reduce p-AKT levels. 

In line with our findings, Mig-6 was shown to be upregulated during the oncogene-induced senescence process and overexpressing of Mig-6 enhances cellular senescence levels in human embryonic lung diploid fibroblast (2BS cells), whereas knockdown of Mig-6 delayed the initiation of Ras-induced cellular senescence [28,29]. Also, it was shown that overexpression of Mig-6 is sufficient to trigger premature cellular senescence of early passage of human diploid lung fibroblasts (WI-38 cells). Notably, the induction of *ERRFI1* mRNA and recruitment of AR by SAL was recently also described for VCaP cells [30]. Furthermore, SAL was shown to enhance *ERRFI1* mRNA levels in patient-derived mouse xenografts treated with vehicle or high-dose androgen [31] (Appendix A). Based on the high degree of tumor heterogeneity of PCa [27], a generalized conclusion cannot be drawn. Therefore, further experiments using different PCa cell lines have to be performed to get more insights into the pathways activated by SAL treatment using in the BAT therapy. 

Taken together, these results suggest that MIG6 induced by genomic AR signaling regulates AKT non-genomic AR signaling in LNCaP cells thus integrating the genomic and non-genomic androgen response.

## Figures and Tables

**Figure 1 biomolecules-12-01048-f001:**
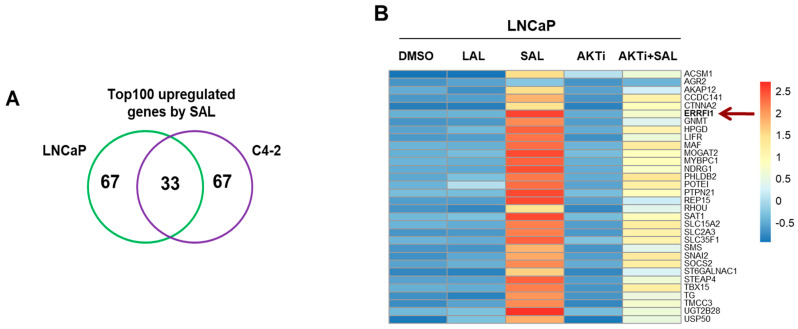
*ERRFI1* expression is upregulated by SAL and inhibited by cotreatment with AKTi. Transcriptome analysis using RNA-seq were performed after treatment of cells for 72 h (n = 3). (**A**) Venn diagram indicates the overlap of the top100 significantly SAL-upregulated genes and their overlap between LNCaP and C4-2 cells. (**B**) Heat map represents the log2 fold change of the 33 genes upregulated upon SAL in LNCaP cells treated with DMSO as a solvent control, low androgen level (LAL, 1 pM R1881), supraphysiological androgen level (SAL, 1 nM R1881) and AKTi (1 μM). Color key number represents normalized count. (**C**–**E**) Normalized log2 fold change of *ERRFI1* upon SAL (**C**), AKTi+SAL (**D**) or AKTi alone (**E**) in both LNCaP and C4-2 cells of RNA-seq data.

**Figure 2 biomolecules-12-01048-f002:**
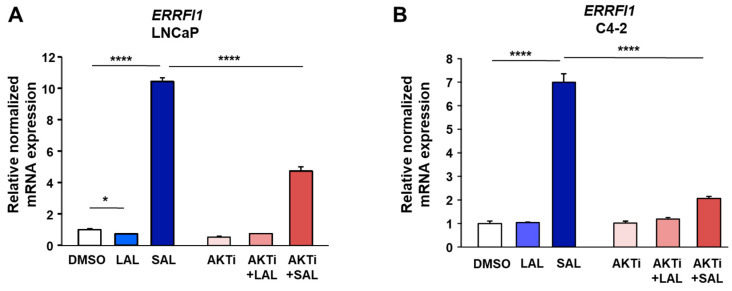
SAL treatment increases, whereas cotreatment with AKTi reduces *ERRFI1* mRNA expression and MIG6 protein levels in both LNCaP and C4-2 cells. Cells were treated as indicated for 72 h. (**A**,**B**) Confirmed regulation of the *ERRFI1* using qRT-PCR in both LNCaP (**A**) and C4-2 cells (**B**). (**C**) Protein extracts of cells treated as indicated were used to perform Western blotting for MIG6 detection. The numbers indicate fold change of the band intensities compared to the DMSO control. (**D**) Chromatin immuno-precipitation with subsequent massive parallel sequencing (ChIP-seq) revealed recruitment of AR to the *ERRFI1* locus.

**Figure 3 biomolecules-12-01048-f003:**
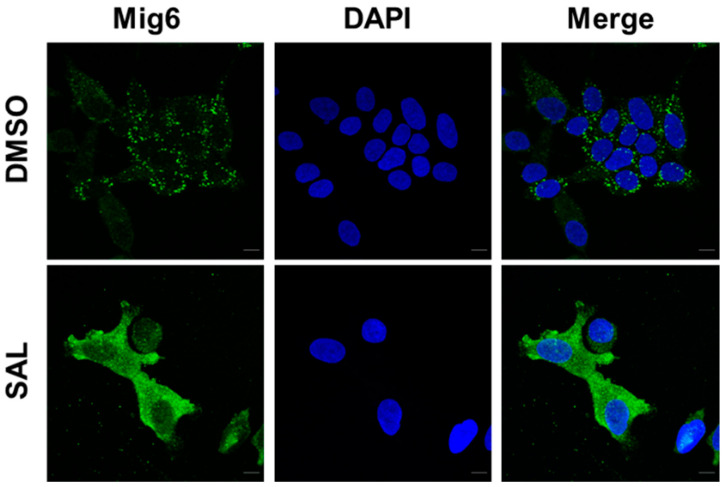
MIG6 levels are enhanced in the cytoplasm upon induction by SAL. Immunofluorescence detection to visualize intracellular localization of MIG6 (green) in LNCaP cells by high-resolution confocal scanning fluorescence microscopy. Nuclei are stained by DAPI (blue). LNCaP cells were treated with DMSO as solvent control or SAL in medium containing 5% FBS for 72 h. DAPI staining was used for staining nuclei. Immunofluorescence using anti-MIG6 antibody, indicates enhanced protein levels being predominantly localized in cytoplasm.

**Figure 4 biomolecules-12-01048-f004:**
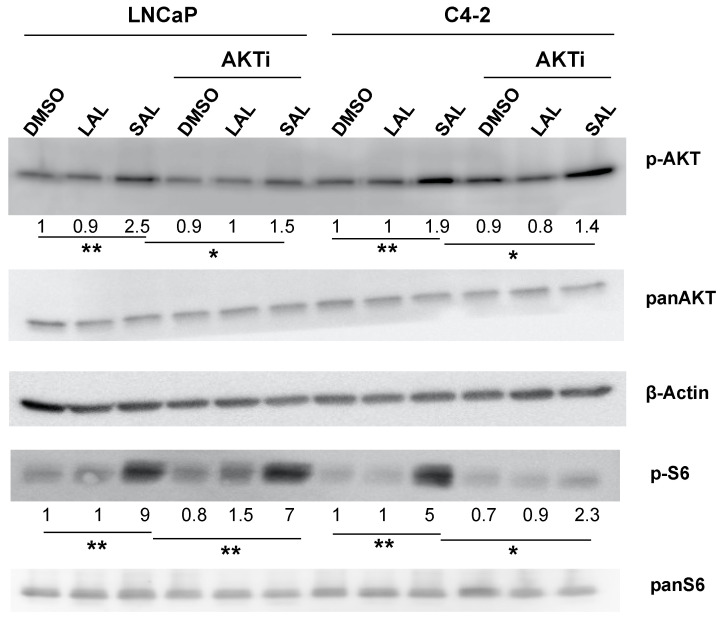
AKTi treatment counteracts the SAL induced p-AKT and p-S6 levels. LNCaP and C4-2 cells were treated with DMSO as solvent control, LAL or SAL with and without AKTi for 72 h. Protein extracts were used for Western blot for detection of p-AKT and pS6. Numbers below the bands indicate the intensities normalized to β-Actin bands as loading control with statistical significance (n = 3).

**Figure 5 biomolecules-12-01048-f005:**
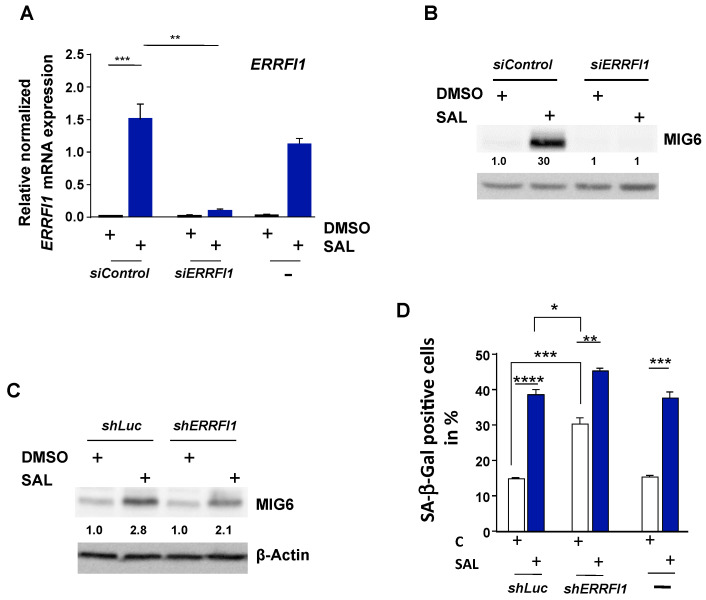
Knockdown of *ERRFI1* significantly decreases *ERRFI1* mRNA and protein expression. LNCaP cells were treated for 72 h with SAL or DMSO as solvent control. RNA and proteins were extracted and analyzed by (**A**) qRT-PCR and Western blotting (**B**,**C**), respectively. (**B**) The *ERRFI1* mRNA expression was normalized to the housekeeper mRNAs *TBP* and *GAPDH*. The mean and SEM values were calculated from 3 biological replicates. The protein levels of MIG6 upon si-mediated knockdown (**B**) or sh-mediated knockdown (**C**) were normalized to the loading control β-Actin using LabImage 1D software (Kapelan Bio Imaging solutions, Leipzig, Germany, Version L340). Numbers indicate a fold-change of protein expression. (**D**) Knockdown of *ERRFI1* induces cellular senescence. The percentage of the SA-β-Gal positive LNCaP cells was calculated in relation to total number of cells per observed field of each transfection and treatment. Two random fields per well were counted, and the mean was calculated.

**Figure 6 biomolecules-12-01048-f006:**
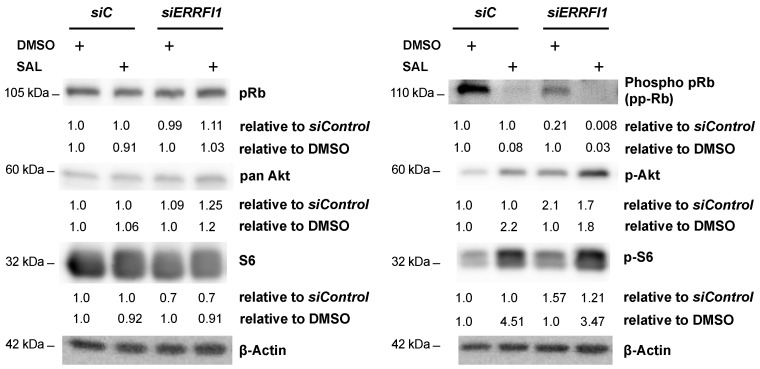
siRNA mediated knockdown of *ERRFI1* decreases phospho-p-Rb (pp-Rb) and enhances p-AKT and p-S6 levels. *siERRFI1* knockdown LNCaP cells treated for 72 h were analyzed for changes in pRb, AKT and S6 levels using Western blotting. siControl (*siC*) served as transfection control. Upper numbers indicate the fold-change of protein level relative to the particular treatment in transfected control cells. Lower numbers indicate a fold-change of protein levels relative to solvent control DMSO.

**Figure 7 biomolecules-12-01048-f007:**
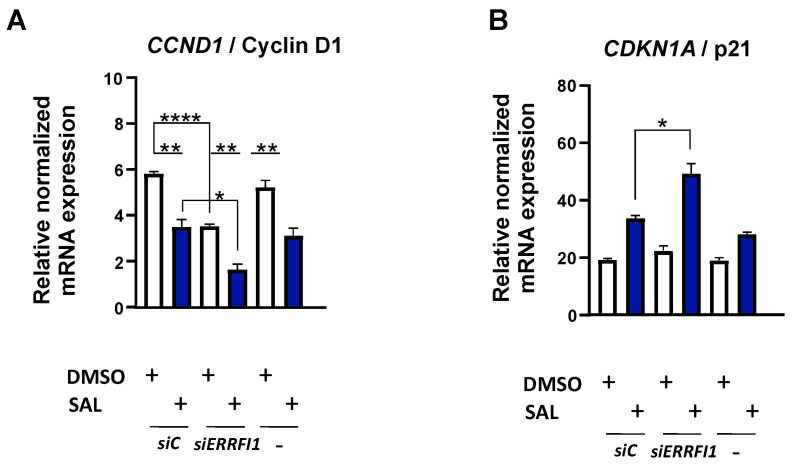
siRNA mediated knockdown of *ERRFI1* enhances levels of cell cycle inhibitors. (**A**–**C**) qRT-PCR were performed for analysis of mRNA expression of *CCND1* (**A**), *CDKN1A* (**B**) and *CDKN2B* (**C**). The expression was normalized to housekeeper mRNAs of *TBP* and *GAPDH*. The mean and SEM values were calculated from 3 biological replicates. (**D**) The protein expression of p15^INK4b^ was normalized to the loading control β-actin using LabImage 1D software.

**Table 1 biomolecules-12-01048-t001:** Primers (5’ to 3’) used for qRT-PCR.

*CCND1* (Cyclin D1)	fwd rev	TCAACCTAAGTTCGGTTCCGATG GTCAGCCTCCACACTCTTGC
*ERRFI1* (MIG6)	fwd rev	GAAGACCTACTGGAGCAGTCG GACTTTTGAGATGGACCATTTTCTG
*FKBP5*	fwd rev	GAGGAAACGCCGATGATTGGAGAC CATGCCTTGATGACTTGGCCTTTG
*GAPDH*	fwd rev	AGTCCCTGCCACACTCAG TACTTTATTGATGGTACATGACAAGG
*CDKN1A* (p21)	fwd rev	TCGACTTTGTCACCGAGACACCAC T CAGGTCCACATGGTCTTCCTCTG
*CDKN2B* (p15^INK4B^)	fwd rev	GATGCGTTCACTCCAATGTC CTTTGTCCTCAGTCTTCAGGTT
*TBP*	fwd rev	GGCGTGTGAAGATAACCCAAGG CGCTGGAACTCGTCTCACT

## Data Availability

The RNA-sequencing data is available in the gene expression omnibus (GEO) database under the accession numbers GSE162711, GSE155528, GSE154755.

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
