# Peer review of "Androgen-Induced MIG6 Regulates Phosphorylation of Retinoblastoma Protein and AKT to Counteract Non-Genomic AR Signaling in Prostate Cancer Cells"

_biomolecules, 2022, doi:10.3390/biom12081048_

Round 1
Reviewer 1 Report
This manuscript focuses on the investigation of the effects of SAL on prostate cancer and crosstalk of AR and AKT signaling upon ASL treatment. The authors identified ERRFl1 as a gene upregulated by SAL and downregulated by AKT and concluded that ERRFl1/MIG6 works in a negative feedback loop to attenuate the decrease in AKT phosphorylation upon SAL.
There are numerous points that should be addressed prior to consideration for publication
Major comments
While some of the results are conclusive, some of the results are not and the biological significance of multiple results should be addressed.
The results demonstrating upregulation of ERRFl1 in two cell lines upon SAL treatment are of interest to increase our understanding of SAL effects. It is surprising that the results presented show significantly increased RNA expression in LNCaP and C4-2 cells, (figure 1) but figure 2 does not show any protein expression in C4-2 cells despite similar ERRFl1 increases shown by RT-PCR.
While RT-PCR results show attenuation of ERRFl1 increase upon SAL after AKT inhibition (Figure 2 A and B), the effects of AKTi on the MIG6 increase upon SAL treatment shown in figure 2C do not really correspond to the results described in the text. It does not appear that AKTi decreased MIG6 levels in LNCaP cells, and moreover no MIG6 was detected in C4-2 cells under any conditions.
Similarly, the results of the effects of SAL and AKTi on phosphorylation of AKT shown in figure 4 are not fully conclusive. SAL increased AKT phosphorylation, which was then decreased by AKTi in LNCaP cells, but rather minimally affected by AKTi in C4-2cells. There are numbers under the gel presented but it is not clear how these were generated, if the * means significance, how many times were these experiments repeated, and moreover, it does not seem that the P-AKT, pan AKT, and b actin were run on the same gel based on the bands' position. It is also surprising that p-AKT levels were not decreased by AKTi under LAL condition, as LNCAP and C4-2 cells are PTEN negative resulting in constitutively active AKT.
Data presented in figure 6B are somewhat difficult to understand. The authors described these results “that higher basal SA-β-Gal positive cells compared to control transfected cells (Fig. 6A, B), indicate that MIG6 protects PCa cells to undergo cellular senescence” and “SAL treatment induces cellular senescence with a slightly but significantly higher percentage of SA-β-Gal positive cells in ERRFI1-knockdown cells was observed. It should be taken into account that the fold-change of SA-β-Gal positive cells in the ERRFI1-knock down cells was reduced. This may indicate that ERRFI1 is involved in protecting these cancer cells to undergo cellular senescence by in SAL.” It is surprising that knockdown of ERRFl1 by shRNA (only ~ 30% knockdown) caused such a large increase in the number of SA-Gal-positive cells since the endogenous expression of this gene was extremely low to start with (figure 1 C indicate that the ERRFl1 expression in untreated cells might have been below the background levels.) Moreover, based on the results presented one could also conclude that the knockdown of ERRFl1 caused attenuation of the increased number of SA-Gal positive cells when compared to increases upon SAL treatment in shLuc cells, and therefore MIG6 is involved in the SAL inhibition of tumor progression. While phosphorylation of Rb in knockdown cells provides support to the authors' conclusion, the increased P-AKT and increased P-S6 upon ERRFl1 knockdown when compared to control cells is more concordant with increases of these upon SAL treatment. Additional experiments, such as the use of additional cell lines, evaluation effects on proliferation under the same conditions, and overexpression of ERRFl1 experiment would be needed to more significantly support the authors’ concussions.
Unfortunately, only LNCaP cell line and its subline were used in this manuscript. Therefore, it is not possible to determine whether these effects are LNCaP specific or present in additional PC models. There are RNASeq data publicly available on multiple cell lines treated with SAL, and analysis of these would address this issue.
Other comments
- Figure 6A is not a publication quality
- The article needs English and general editing
“Taken together, these data indicate that MIG6 regulates the key factors associated
272 with and mediating induction of cellular senescence by SAL treatment. This section may
273 be divided by subheadings. It should provide a concise and precise description of the ex-
274perimental results, their interpretation, as well as the experimental conclusions that can
275be drawn.
“Figure 4. AKTi treatment counteracts the levels of SAL induced p-AKT and p-S6 levels. LNCaP and
209 C4-2 cells were treated with DMSO (D) as solvent control, LAL (L), or SAL (S) with and without
210 AKTi for 72 hours. Protein extracts were used for Western blot for detection of p-AKT and pS6.
211 Numbers below the bands indicate the intensities normalized to β-Actin bands as loading control.” The abbreviations L, D, etc are not used in the figure and not needed,
Author Response
We sincerely thank the reviewers for their support to enhance the value of our manuscript.
We dealt with the reviewer’s comments as follows:
Reviewer 1:
Thank you for rating our manuscript as interest and that it leads to increase our understanding of SAL effects.
The results demonstrating upregulation of ERRFl1 in two cell lines upon SAL treatment are of interest to increase our understanding of SAL effects. It is surprising that the results presented show significantly increased RNA expression in LNCaP and C4-2 cells, (figure 1) but figure 2 does not show any protein expression in C4-2 cells despite similar ERRFl1 increases shown by RT-PCR. While RT-PCR results show attenuation of ERRFl1 increase upon SAL after AKT inhibition (Figure 2 A and B), the effects of AKTi on the MIG6 increase upon SAL treatment shown in figure 2C do not really correspond to the results described in the text. It does not appear that AKTi decreased MIG6 levels in LNCaP cells, and moreover no MIG6 was detected in C4-2 cells under any conditions.
Response: The blots are now shown with densitometric analyses. Indeed, for C4-2 cells the MIG6 bands are too week since we loaded less protein extracts. Now the blot is shown with higher level of beta-Actin by loading more extracts to analyze better the MIG6 protein levels.
Treatment with AKTi reduces MIG6 protein levels in both cell lines. See revised text and Fig. 2.
Similarly, the results of the effects of SAL and AKTi on phosphorylation of AKT shown in figure 4 are not fully conclusive. SAL increased AKT phosphorylation, which was then decreased by AKTi in LNCaP cells, but rather minimally affected by AKTi in C4-2cells.
Response: We do see inhibition of p-S6 levels in a similar to LNCaP cells indicating that AKTi inhibits AKT-signaling also in C4-2 cells. Indeed, we and other have shown that AR enhances levels of phosphorylated AKT in LNCaP and castration resistant C4-2 cells (Roediger et al., 2014, Mirzakhani et al., 2021), which could be a basis for castration resistance. The reduction of phospho-AKT levels by AKTi are indeed less pronounced as in C4-2 cells but statistically significant (n=3). This might be due to the castration-resistance of C4-2 cells.
We now discuss this on page 7.
There are numbers under the gel presented but it is not clear how these were generated, if the * means significance, how many times were these experiments repeated, and moreover, it does not seem that the P-AKT, pan AKT, and b actin were run on the same gel based on the bands' position.
Response: We have repeated the Western blots three times (n=3), which is now indicated in the figure legend. In the supplement we show the membranes with the corresponding beta-Actin loading control. The stars mean significant as indicated in material and methods.
It is also surprising that p-AKT levels were not decreased by AKTi under LAL condition, as LNCAP and C4-2 cells are PTEN negative resulting in constitutively active AKT.
Response: We repeatedly observe that LAL does not promote a further enhancement of AKT phosphorylation (Roediger et al., 2014). We also observed consistently reduction of p-AKT by AKTi. Phospho-AKT has a basal level in these cells, presumably by PTEN mutation, which however, is enhanced specifically by SAL treatment.
Data presented in figure 6B are somewhat difficult to understand. The authors described these results "that higher basal SA-β-Gal positive cells compared to control transfected cells (Fig. 6A, B), indicate that MIG6 protects PCa cells to undergo cellular senescence" and "SAL treatment induces cellular senescence with a slightly but significantly higher percentage of SA-β-Gal positive cells in ERRFI1-knockdown cells was observed.
It is surprising that knockdown of ERRFl1 by shRNA (only ~ 30% knockdown) caused such a large increase in the number of SA-Gal-positive cells since the endogenous expression of this gene was extremely low to start with (figure 1 C indicate that the ERRFl1 expression in untreated cells might have been below the background levels.)
Response: Fig. 1C shows the fold induction of RNA-seq data with the padj values. This does not necessarily indicate the background levels.
We have performed SA-beta gal activity staining in addition to sh-vector-mediated knockdown also knockdown by si-RNA and observed a similar increase in cellular senescence levels confirming the observation that MIG6 knockdown itself enhances cellular senescence levels confirming the data and suggest that a knockdown with different efficacy leads to enhanced cell senescence levels. See new supplementary data Fig. S5.
Moreover, based on the results presented one could also conclude that the knockdown of ERRFl1 caused attenuation of the increased number of SA-Gal positive cells when compared to increases upon SAL treatment in shLuc cells, and therefore MIG6 is involved in the SAL inhibition of tumor progression.
Response: This is indeed one possible interpretation of the results which we emphasized now in discussion, see page 11.
While phosphorylation of Rb in knockdown cells provides support to the authors' conclusion, the increased P-AKT and increased P-S6 upon ERRFl1 knockdown when compared to control cells is more concordant with increases of these upon SAL treatment. Additional experiments, such as the use of additional cell lines, evaluation effects on proliferation under the same conditions, and overexpression of ERRFl1 experiment would be needed to more significantly support the authors' concussions.
Response: This work compares and focusses on the AR-SAL-mediated change of expression of MIG6 in two cell AR expressing established lines. The focus is to analyze the SAL-mediated cellular senescence. Therefore, we used knockdown since SAL enhances the MIG6 expression.
Indeed, increase in these phospho levels by SAL is in line with higher levels of cellular senescence and supports our observation that p-AKT is a mediator of SAL-induced cellular senescence. Our hypothesis is that SAL increases p-AKT levels and enhances a pro-survival pathway to induce cellular senescence (Pungsrinont et al., 2020; Kokal et al., 2021). We now discussed this point. See page 11.
Unfortunately, only LNCaP cell line and its subline were used in this manuscript. Therefore, it is not possible to determine whether these effects are LNCaP specific or present in additional PC models. There are RNASeq data publicly available on multiple cell lines treated with SAL, and analysis of these would address this issue.
Response: Indeed, the recent contribution by Basil et al. confirms upregulation of ERRFl1 in VCaP cells by SAL (Basil et al., Scientific reports, 2022). This is now included in page 11.
Other comments
Figure 6A is not a publication quality. Has now been deleted
The article needs English and general editing. We edited the ext.
"Taken together, these data indicate that MIG6 regulates the key factors associated
272 with and mediating induction of cellular senescence by SAL treatment. This section may
273 be divided by subheadings. It should provide a concise and precise description of the ex-
274perimental results, their interpretation, as well as the experimental conclusions that can
275be drawn.
Response: We now divided into two paragraphs.
"Figure 4. AKTi treatment counteracts the levels of SAL induced p-AKT and p-S6 levels. LNCaP and
209 C4-2 cells were treated with DMSO (D) as solvent control, LAL (L), or SAL (S) with and without
210 AKTi for 72 hours. Protein extracts were used for Western blot for detection of p-AKT and pS6.
211 Numbers below the bands indicate the intensities normalized to β-Actin bands as loading control." The abbreviations L, D, etc are not used in the figure and not needed,
Response: We have now deleted these abbreviations.
Reviewer 2 Report
In this manuscript, Schomann et al. performed RNAseq on two prostate cancer (PC) cell lines treated with supraphysiological androgen levels (SAL), and they discovered that SAL upregulates MIG6 to induce PC cell senescence. Unfortunately the manuscript currently suffers from poor data quality (please see details below). Most importantly, it doesn’t make sense at all that SAL induces MIG6 expression and senescence, whereas knocking down MIG6 also induced senescence and did not reverse/rescue SAL-induced senescence. What I can conclude from the data presented so far is that the induction of MIG6 doesn’t contribute to SAL-induced senescence at all. As such, substantial experimental work would be needed before it can reach to a satisfactory level for publication.
Other major issues:
- In general, the authors only looked at one single time point and a single SAL/low androgen level (LAL) dose. This clearly falls short for a rigorous study. The authors should explore a range of time points and androgen doses to fully elucidate changes in MIG6 mRNA and protein levels in response to androgen.
- Figure 2C: AKTi doesn’t seem to reduce MIG6 protein levels induced by SAL, do the authors have an explanation to that? This experiment needs to be done for at least 3 times and MIG6:beta-actin ratio needs to be quantified. MIG6 bands on C4-2 are weak but still should be visible upon a higher exposure level, and it’d be important to show that. The authors can show the blots separately if needed.
- Figure 4: Although changes in P-S6 is quite clear, but stats for n = 3 should be shown for P-AKT instead of n = 1, because the differences were quite subtle from the only presented n.
- Figure 5C: Again, this is only n = 1 and according to the WB it seems like the shRNA is not working properly. For how long have the cells been transfected with the shRNA? I’d suggest doing a time course experiment (48, 72, 96 and 120 h) post-shRNA transfection to determine the level of MIG6 protein changes over time. Also, if the siRNA was perfect (Figure 5B), why need to try shRNA?
- Figure 6: It is necessary to show that physiological level of androgen (LAL?) doesn’t induce cell senescence.
- Why out of a sudden the authors want to look at senescence? There were previous reports talking about how MIG6 protein levels are upregulated in senescent cells, if this was the reason, the authors should at least explain it and cite those papers.
- Figure 7. Are the data all from LNCaPs? Especially for panel A, what’s the difference between the left and the right? Are they from different PC cell lines? If not why there are two actin blots for a single experiment? If all P-blots are meant to be placed on the left/right-hand side then this needs to be consistent.
- Figure 7D: The most important stats to show here is whether it is statistically/significantly different between the 2nd and the 4th It looks like there is, if yes, please show.
Minor issues:
- Lines 204-205: Should be “pronouncedly reduced”.
- Figure 4: P-S6 label is missing.
Author Response
We sincerely thank the reviewers for their support to enhance the value of our manuscript.
Reviewer 2:
Most importantly, it doesn't make sense at all that SAL induces MIG6 expression and senescence, whereas knocking down MIG6 also induced senescence and did not reverse/rescue SAL-induced senescence. What I can conclude from the data presented so far is that the induction of MIG6 doesn't contribute to SAL-induced senescence at all. As such, substantial experimental work would be needed before it can reach to a satisfactory level for publication.
Response: In general, cells have feed-back loops for many pathways. Changing the level of one factor is likely to induce feed-back loops. SAL changes the expression of many factors. In fact, MIG6 was described as a multi-adaptor and scaffold protein composed of many domains that makes it multifunctional (Anastasi et al., 2014). The increase of cellular senescence by Mig6 knockdown indicates that MIG6 regulates cellular senescence and induces a pro-survival pathway. This is in line with reduced p-Rb, and enhanced p-AKT and p-S6 levels of the non-genomic AR pathway. It also suggests that MIG6 protects PCa cells to undergo cellular senescence. We have emphasized this point. See Page 11.
Other major issues:
In general, the authors only looked at one single time point and a single SAL/low androgen level (LAL) dose. This clearly falls short for a rigorous study. The authors should explore a range of time points and androgen doses to fully elucidate changes in MIG6 mRNA and protein levels in response to androgen.
Response: We have added now a time-dependent and concentration dependent analysis of MIG6 expression, see new supplemental Fig. S3
Figure 2C: AKTi doesn't seem to reduce MIG6 protein levels induced by SAL, do the authors have an explanation to that? This experiment needs to be done for at least 3 times and MIG6:beta-actin ratio needs to be quantified. MIG6 bands on C4-2 are weak but still should be visible upon a higher exposure level, and it'd be important to show that. The authors can show the blots separately if needed.
Response: The blots are now shown with densitometric analyses. Indeed, for C4-2 cells the MIG6 bands are too week since we loaded less protein extracts. Now the blot is shown with higher level of beta-Actin by loading more extracts to analyze better the MIG6 protein levels.
Treatment with AKTi reduces MIG6 protein levels in both cell lines. See revised text and Fig. 2.
Figure 4: Although changes in P-S6 is quite clear, but stats for n = 3 should be shown for P-AKT instead of n = 1, because the differences were quite subtle from the only presented n.
Response: Change of p-AKT by SAL were repeatedly shown in many experiments. We added now the stats for three independent experiments in the supplementary file.
For how long have the cells been transfected with the shRNA? I'd suggest doing a time course experiment (48, 72, 96 and 120 h) post-shRNA transfection to determine the level of MIG6 protein changes over time. Also, if the siRNA was perfect (Figure 5B), why need to try shRNA?
Response: The use sh-vectors and stable knockdown for longterm culturing, was not more efficient. Actually, cells loose rapidly the sh-mediated knockdown. Surprisingly, the si knockdown was more efficient. Our hypothesis is that a potent knockdown might induce cell stress and senescence as well as cell detachment, which we have some indications. Now we show also for si-mediated knockdown of MIG6, cells respond, similar to sh-MIG6, by an increased level of cellular senescence confirming the data. See new Figure S5.
Figure 6: It is necessary to show that physiological level of androgen (LAL?) doesn't induce cell senescence.
Response: We had shown previously that LAL does neither enhance phospho-RB nor p-AKT levels (Roediger et al., 2014; Mirzakhani et al., 2021).
Why out of a sudden the authors want to look at senescence? There were previous reports talking about how MIG6 protein levels are upregulated in senescent cells, if this was the reason, the authors should at least explain it and cite those papers.
Response: We now cited previous contributions that confirm MIG6 is involved in induction of cellular senescence. See page 12 of discussion.
Based on our observation that SAL induces cellular senescence in these PCa cells and MIG6 being upregulated, MIG6 was further analyzed. Interestingly, MIG6 regulates the same pro-survival pathway (p-AKT, p-S6 and the pRb pathway) as the AR using SAL. Response: We have now enhanced this part and the rational to analyze senescence.
We added now the rationale in more detail why cellular senescence was analyzed. See pages 2, 8 and 9.
Figure 7. What's the difference between the left and the right? Are they from different PC cell lines? If not why there are two actin blots for a single experiment? If all P-blots are meant to be placed on the left/right-hand side then this needs to be consistent.
Response: On the left side the levels of pan-AKT, pan-S6 and pRb levels are shown, whereas on the right side the phosphorylated AKT and S6 as well as the hyperphosphorylation of pRb (ppRb) are indicated. For each blot a individual Actin detection is required since the phosphorylated and unphosphorylated proteins can not be detected on the same blot. For additional controls for Western blots please see the supplementary data.
Figure 7D: The most important stats to show here is whether it is statistically/significantly different between the 2nd and the 4th It looks like there is, if yes, please show.
Response: Yes, relative to siControl the p15 increase is statistically significant.
Minor issues:
Lines 204-205: Should be "pronouncedly reduced".
Response: Thank you. We revised the text.
Figure 4: P-S6 label is missin
Response: Thank you. We added p-S6
Reviewer 3 Report
In the present manuscript from Schomann and colleagues, the group analyses the role of MIG6 in the non-genomic AR signaling pathway. Thereby the group describes the regulation of MIG6 by supraphysiological androgen levels. The study sounds interesting and introduces a novel role of MIG6 in androgen receptor signaling in prostate cancer. However, several issues need to be handled before publishing:
Introduction:
- According to the latest guidelines, only advanced and metastasized PCa are recommended to be treated with androgen deprivation therapy (ADT) or ADT combined with Antiandrogens/Chemotherapy. This recommendation should be mentioned in the introduction. In addition, the latest guidelines should be cited and not a paper from 2012.
- Supraphysiological androgen levels, as well as bipolar androgen therapy, should be defined in detail. Readers from different fields may not understand the meaning.
- As R1881 and dihydrotestosterone are compared in the introduction, the two molecules should be briefly introduced and their differences explained.
- Line 61: Reference Roediger et al. 2014 seems not to be correctly implemented.
- The non-genomic and genomic AR pathways should be briefly introduced as they are a central topic of the manuscript.
Methods:
- The cell line choice should be explained. Moreover, a short overview of the main characteristics of the cell lines would be desirable.
- 10 µM Bic and 10 µM ENZ are not used in this manuscript
- Even if cited correctly, a brief description of the RNA sequencing experiment should be given. For example, treatment time and treatment concentrations.
- Lot Number of the antibodies should be added
Results+Discussion:
- Are the 100 presented genes all unique for SAL?
- Cells should also be treated with antiandrogens to validate the genomic AR pathway. Moreover, as high ligand concentrations can lead to unspecific binding androgens to other steroid receptors, siAR experiments should be analyzed. ChIP seq analysis of the ERRFI1 gene from previously published studies would strengthen the presented data.
- Displaying the densitometry as column bar graphs with standard deviation would strengthen their findings. Moreover, significant western blot differences should be highlighted more clearly.
- It is not clear to this reviewer how to hypothesize the changed protein phosphorylation by RNAseq data (line 218). However, the influence of MIG6 was demonstrated in other studies earlier. Therefore, the hypothesis must be explained, or the statement must be cited correctly.
- At least the main senescence experiments must also be done in C4-2 cells. Therefore, it would be investigated if it is a general mechanism or specific for castration-sensitive cells.
- Please revise lines 273-276
- The main conclusion is confusing and should be rewritten (lines 322-324).
Author Response
We sincerely thank the reviewers for their support to enhance the value of our manuscript.
Reviewer 3:
Thank you for rating the study being interesting and introduces a novel role of MIG6 in androgen receptor signaling in prostate cancer.
Introduction:
According to the latest guidelines, only advanced and metastasized PCa are recommended to be treated with androgen deprivation therapy (ADT) or ADT combined with Antiandrogens/Chemotherapy. This recommendation should be mentioned in the introduction. In addition, the latest guidelines should be cited and not a paper from 2012.
Response: Thank you very much pointing this out. We have changed and replaced with a new guideline, see citation #3.
Supraphysiological androgen levels, as well as bipolar androgen therapy, should be defined in detail. Readers from different fields may not understand the meaning.
Response: That is an important point. The exact concentrations were provided in the introduction (line 58-60), figure legend of Fig. 1, and is now in addition indicated in Material methods lane 89 and supplemental Fig. S2. Further BAT and the use of SAL is now introduced in more detail, see page 2.
As R1881 and dihydrotestosterone are compared in the introduction, the two molecules should be briefly introduced and their differences explained.
Response: We explained now the higher AR-specificity for the AR agonist R1881 and added this information on page 2
Line 61: Reference Roediger et al. 2014 seems not to be correctly implemented.
Response: Thank you very much. We have changed it to [8].
The non-genomic and genomic AR pathways should be briefly introduced as they are a central topic of the manuscript.
Response: Thank you very much pointing this out. We have emphasized this point. See page 2
Methods:
The cell line choice should be explained. Moreover, a short overview of the main characteristics of the cell lines would be desirable.
Response: We added information on page 3.
10 µM Bic and 10 µM ENZ are not used in this manuscript
Response: Both BIC and Enz were used to reveal AR ligand specific regulation of MIG6, shown in the supplementary Figure S2.
Even if cited correctly, a brief description of the RNA sequencing experiment should be given. For example, treatment time and treatment concentrations.
Response: We added this information now on page 3
Lot Number of the antibodies should be added
Response: We added information on page 3.
Results+Discussion:
Are the 100 presented genes all unique for SAL?
Response: The Top100 upregulated genes are those genes that showed the highest induction by SAL in each cell line.
Cells should also be treated with antiandrogens to validate the genomic AR pathway. Moreover, as high ligand concentrations can lead to unspecific binding androgens to other steroid receptors, siAR experiments should be analyzed. ChIP seq analysis of the ERRFI1 gene from previously published studies would strengthen the presented data.
Response: We have included the treatment of two AR antagonists Enzalutamide and Bicalutamide and confirmed that ERRFI1 expression is induced by SAL and not by AR antagonists.
We had shown previously that the induction of cellular senescence is AR dependent. PC3 cells lacking AR did not whereas PC3-AR cells showed induction by SAL:
We had used previously siAR knockdown which however resulted in apoptosis of cells which is in agreement with the observations of Kim et al. (2013) although we are aware that other groups were able to knockdown AR in their sublines.
Concerning ChIP experiments, we performed ChIP-seq experiments. ERRF1 is in the statistically significant list and recruits the AR to the ERRF1 Gene. We used IGV software and added the location in the new Fig. 2D.
Displaying the densitometry as column bar graphs with standard deviation would strengthen their findings. Moreover, significant western blot differences should be highlighted more clearly.
Response: We added stars for statistically significant changes. In addition, we added in supplement the statistics, see supplemental Western blot files.
It is not clear to this reviewer how to hypothesize the changed protein phosphorylation by RNAseq data (line 218). However, the influence of MIG6 was demonstrated in other studies earlier. Therefore, the hypothesis must be explained, or the statement must be cited correctly.
Response: Thank you for pointing this out. We have changed the statements. See page 8
At least the main senescence experiments must also be done in C4-2 cells. Therefore, it would be investigated if it is a general mechanism or specific for castration-sensitive cells.
Response: SAL-induced cell senescence and RNA-seq was performed in both LNCaP and C4-2 cells identifying MIG6 being upregulated in both cell lines. However, we did not intent to make a generalize statement. Especially in the context of a high degree of tumor heterogenicity and plasticity of prostate cancer. We mentioned this restriction now on page 12.
Please revise lines 273-276
Response: Thank you. We revised the text.
The main conclusion is confusing and should be rewritten (lines 322-324).
Response: Thank you. We revised the text.
Round 2
Reviewer 1 Report
My issues with the quality and interpretation of the results have not been resolved.
1) For example, the text states: “One of the prominently induced genes is ERRFI1, encoding MIG6, being upregulated by SAL and downregulated by AKTi in the cotreatment with SAL, whereas LAL did not up-regulate ERRFI1 expression (Fig. 1B, C, D). In figure D the decrease in EREFI1 has adjusted p0.8 (LNCaP) and 0.99 (C4-2). With p values like this- one cannot state thattheERRFI1 was decreased by SAL + AKTi. While the decreased ERRFI1 was shown in figure 2A, it is unclear how was the ERRFI1 chosen for studies of AKTi on SAL since the RNA-Seq did not show any significant changes.
2) authors state:” Time-dependent androgen treatment indicates that MIG6 protein levels are enhanced as an early response, decline after 1hour and are induced by 48 and 72 hours treatment with SAL (Supplementary Fig. S3).” The quality of WB in figure S3 does not fully support the claim of upregulation in one hour
3) SAL enhances in both cell lines the levels of p-AKT and p-S6 (Fig. 4) This statement is supported by the WB data. However, it is rather puzzling, and potentially indicates a technical issue, that AKTi did not alter levels of P-AKT in either cell line really (LNCaP: 1-0.9, and C4-2: 1-1). These cell lines have constitutively active AKT signaling, and the AKT inhibitor should decrease the basal levels of P- as well as P-S6.
4) The authors show data that shERRFI1 increases senescence under normal well as SAL conditions (Fig 5D) authors state supporting data show similar results in supplemental figure 5. However, supplemental figure 5 does not show the significance of the difference in senescent cells number with siERRFI1 under SAL conditions nor the significant difference between senescent cell numbers with and with SAL in siERRFI1 KD cells.
Figure 3 shows cytoplasmic localization of MIG6 under low/no androgen and SAL, indicating cytoplasmic localization of MIG6. The authors introduce an article that shows AR effects on MIG6 localization in cytoplasm vs nucleus. This issue should be addressed since no nuclear immunoreactivity was present in figure 3.
The original concern that the results are solely specific to LNCaP and its subline C4-2 cells was not addressed appropriately. In response, the authors stated in the revised manuscript: ”Based on the high degree of tumor heterogeneity of PCa [27] a generalized conclusion cannot be drawn. Therefore, further experiments using different PCa cell lines have to be performed to get more insights into the pathways activated by SAL treatment…” There are RNASeq data sets available for PC cell lines and PDX treated with SAL, the authors should at least interrogate the existing datasets whether the mechanism shown in LNCaP cells is taking place in other PC models. ”
Some of the newly added language duplicated the original text E.g., With 49
BAT treatment, PCa patients receive intermittent androgen injections at doses shown to 50
produce a spike in serum androgens to supraphysiological levels, followed by a decline 51
to below castrate levels at the end of a 28-day treatment cycle. Rapid cycling of androgens 52
from SAL levels to below normal levels may delay adaptive changes in AR signaling and 53
thereby delaying the emergence of resistance [12]. Additionally, an intramuscular depot 54
injection of testosterone was used to induce SAL for the first few days. In the following 55
month the androgen level drops back down to approximately castrate range.
Another additional text states “non-genomic activity of AR” in phosphorylation of AKT. As AR is not kinase it cannot phosphorylate AKT and therefore this activity is still transcriptional regulation of kinases that are involved in phosphorylation of AKT, not non-genomic activity of AR.
”The AR has in addition to its genomic activity as a transcription factor also non-genomic activity. This includes the phosphorylation of AKT and activation of AKT downstream signaling through mTOR and S6 by SAL[8]. “
Author Response
We sincerely thank the reviewer 1 for his support to enhance the value of our manuscript.
Reviewer 1:
- The reviewer raises the point that the RNAseq did not reveal the statistical significance for AKTi-mediated downregulation.
Response: The ERRFI1 was listed within the top100 significantly SAL-upregulated genes and is among the 33top SAL-mediated upregulated genes. We mentioned that now more precisely in lane 198. Indeed, using the RNA-seq dataset no statistical significance for AKTi-mediated downregulation was observed. Therefore qRT-PCR were used for both cell lines. The data suggest a significant downregulation by AKTi (Fig. 2A and B). We pointed that out in lines see page 6.
- authors state:” Time-dependent androgen treatment indicates that MIG6 protein levels are enhanced as an early response, decline after 1hour and are induced by 48 and 72 hours treatment with SAL (Supplementary Fig. S3).” The quality of WB in figure S3 does not fully support the claim of upregulation in one hour.
Response: We tuned down our statement concerning 1 hour treatment. See page 7.
- SAL enhances in both cell lines the levels of p-AKT and p-S6 (Fig. 4) This statement is supported by the WB data. However, it is rather puzzling, and potentially indicates a technical issue, that AKTi did not alter levels of P-AKT in either cell line really (LNCaP: 1-0.9, and C4-2: 1-1). These cell lines have constitutively active AKT signaling, and the AKT inhibitor should decrease the basal levels of P- as well as P-S6.
Response: We assume the reviewer refers to basal level p-AKT and p-S6 with or without AKTi treatment.
Indeed, we observed not much of changes of basal level of p-AKT but a mild reduction of p-S6 levels. Please note, that not all AKT inhibitors inhibit p-AKT levels (Davis et al., 2012; Thomas et al., 2013).
- The authors show data that shERRFI1 increases senescence under normal well as SAL conditions (Fig 5D) authors state supporting data show similar results in supplemental figure 5. However, supplemental figure 5 does not show the significance of the difference in senescent cells number with siERRFI1 under SAL conditions nor the significant difference between senescent cell numbers with and with SAL in siERRFI1 KD cells.
Response: We have now added also the statistics comparing these data.
- Figure 3 shows cytoplasmic localization of MIG6 under low/no androgen and SAL, indicating cytoplasmic localization of MIG6. The authors introduce an article that shows AR effects on MIG6 localization in cytoplasm vs nucleus. This issue should be addressed since no nuclear immunoreactivity was present in figure 3.
Response: We do not want to exclude nuclear localization of MIG6 since we do observe under SAL treatment signals in the nuclei.
- There are RNAseq data sets available for PC cell lines and PDX treated with SAL, the authors should at least interrogate the existing datasets whether the mechanism shown in LNCaP cells is taking place in other PC models.”
Response: We included now RNAseq data from another PCa cell model and included data from patient derived xenograft model system. LuCaP xenografts were used in castrated mice vehicle or high-dose androgen treated. The data support our conclusion that the expression of ERRFI1 is enhanced by high-dose androgen treatment. See lines 436 and 437 as well as supplemental Fig. S6.
- Some of the newly added language duplicated the original text.
Response Thank you for pointing that out. We have now deleted the duplicated part.
- The reviewer mentions that “non-genomic activity of AR” might be due to genomic AR activity regulating kinase expression at genomic level. Indeed, genomic -based regulation might derive from long-term effects of androgen treatment. However, there are evidence in literature that non-genomic AR activity occurs within few minutes of androgen treatment (Peterziel et al., 1999; Liao et al., 2013; Falkenstein et al., 2000), which cannot be easily explained to occur at genomic level.
Reviewer 2 Report
Most importantly, it doesn't make sense at all that SAL induces MIG6 expression and senescence, whereas knocking down MIG6 also induced senescence and did not reverse/rescue SAL-induced senescence. What I can conclude from the data presented so far is that the induction of MIG6 doesn't contribute to SAL-induced senescence at all. As such, substantial experimental work would be needed before it can reach to a satisfactory level for publication.
Response: In general, cells have feed-back loops for many pathways. Changing the level of one factor is likely to induce feed-back loops. SAL changes the expression of many factors. In fact, MIG6 was described as a multi-adaptor and scaffold protein composed of many domains that makes it multifunctional (Anastasi et al., 2014). The increase of cellular senescence by Mig6 knockdown indicates that MIG6 regulates cellular senescence and induces a pro-survival pathway. This is in line with reduced p-Rb, and enhanced p-AKT and p-S6 levels of the non-genomic AR pathway. It also suggests that MIG6 protects PCa cells to undergo cellular senescence. We have emphasized this point. See Page 11.
Reviewer: I am not sure where the authors mentioned about this on “page 11”, maybe because I can only see the edited version of main text + figures so we are talking about a different page number here. Can the authors please specify?
Other major issues:
In general, the authors only looked at one single time point and a single SAL/low androgen level (LAL) dose. This clearly falls short for a rigorous study. The authors should explore a range of time points and androgen doses to fully elucidate changes in MIG6 mRNA and protein levels in response to androgen.
Response: We have added now a time-dependent and concentration dependent analysis of MIG6 expression, see new supplemental Fig. S3
Reviewer: OK.
Figure 2C: AKTi doesn't seem to reduce MIG6 protein levels induced by SAL, do the authors have an explanation to that? This experiment needs to be done for at least 3 times and MIG6:beta-actin ratio needs to be quantified. MIG6 bands on C4-2 are weak but still should be visible upon a higher exposure level, and it'd be important to show that. The authors can show the blots separately if needed.
Response: The blots are now shown with densitometric analyses. Indeed, for C4-2 cells the MIG6 bands are too week since we loaded less protein extracts. Now the blot is shown with higher level of beta-Actin by loading more extracts to analyze better the MIG6 protein levels.
Treatment with AKTi reduces MIG6 protein levels in both cell lines. See revised text and Fig. 2.
Reviewer: OK.
Figure 4: Although changes in P-S6 is quite clear, but stats for n = 3 should be shown for P-AKT instead of n = 1, because the differences were quite subtle from the only presented n.
Response: Change of p-AKT by SAL were repeatedly shown in many experiments. We added now the stats for three independent experiments in the supplementary file.
Reviewer: OK.
For how long have the cells been transfected with the shRNA? I'd suggest doing a time course experiment (48, 72, 96 and 120 h) post-shRNA transfection to determine the level of MIG6 protein changes over time. Also, if the siRNA was perfect (Figure 5B), why need to try shRNA?
Response: The use sh-vectors and stable knockdown for longterm culturing, was not more efficient. Actually, cells loose rapidly the sh-mediated knockdown. Surprisingly, the si knockdown was more efficient. Our hypothesis is that a potent knockdown might induce cell stress and senescence as well as cell detachment, which we have some indications. Now we show also for si-mediated knockdown of MIG6, cells respond, similar to sh-MIG6, by an increased level of cellular senescence confirming the data. See new Figure S5.
Reviewer: OK.
Figure 6: It is necessary to show that physiological level of androgen (LAL?) doesn't induce cell senescence.
Response: We had shown previously that LAL does neither enhance phospho-RB nor p-AKT levels (Roediger et al., 2014; Mirzakhani et al., 2021).
Reviewer: OK. Has this been mentioned in the text? If yes, where?
Why out of a sudden the authors want to look at senescence? There were previous reports talking about how MIG6 protein levels are upregulated in senescent cells, if this was the reason, the authors should at least explain it and cite those papers.
Response: We now cited previous contributions that confirm MIG6 is involved in induction of cellular senescence. See page 12 of discussion.
Based on our observation that SAL induces cellular senescence in these PCa cells and MIG6 being upregulated, MIG6 was further analyzed. Interestingly, MIG6 regulates the same pro-survival pathway (p-AKT, p-S6 and the pRb pathway) as the AR using SAL. Response: We have now enhanced this part and the rational to analyze senescence.
We added now the rationale in more detail why cellular senescence was analyzed. See pages 2, 8 and 9.
Reviewer: OK.
Figure 7. What's the difference between the left and the right? Are they from different PC cell lines? If not why there are two actin blots for a single experiment? If all P-blots are meant to be placed on the left/right-hand side then this needs to be consistent.
Response: On the left side the levels of pan-AKT, pan-S6 and pRb levels are shown, whereas on the right side the phosphorylated AKT and S6 as well as the hyperphosphorylation of pRb (ppRb) are indicated. For each blot a individual Actin detection is required since the phosphorylated and unphosphorylated proteins can not be detected on the same blot. For additional controls for Western blots please see the supplementary data.
Reviewer: OK.
Figure 7D: The most important stats to show here is whether it is statistically/significantly different between the 2nd and the 4th It looks like there is, if yes, please show.
Response: Yes, relative to siControl the p15 increase is statistically significant.
Reviewer: OK.
Minor issues:
Lines 204-205: Should be "pronouncedly reduced".
Response: Thank you. We revised the text.
Reviewer: OK.
Figure 4: P-S6 label is missin
Response: Thank you. We added p-S6
Reviewer: OK.
Author Response
We sincerely thank the reviewer for his support to enhance the value of our manuscript.
- A) 1st Response: It also suggests that MIG6 protects PCa cells to undergo cellular senescence. We have emphasized this point. See Page 11.
Reviewer: I am not sure where the authors mentioned about this on “page 11”, maybe because I can only see the edited version of main text + figures so we are talking about a different page number here. Can the authors please specify?
Response: We have stated this in line 327 on page 10.
- B) 1st Response: We had shown previously that LAL does neither enhance phospho-RB nor p-AKT levels (Roediger et al., 2014; Mirzakhani et al., 2021).
Reviewer: OK. Has this been mentioned in the text? If yes, where?
Response: We stated this now in lines 273 and 274 on page 8.
Reviewer 3 Report
The revised version can be published in its present form.
Author Response
Thank you for the revised version that can be published in its present form.